# A Numerical Investigation of the Mixing Performance in a Y-Junction Microchannel Induced by Acoustic Streaming

**DOI:** 10.3390/mi13020338

**Published:** 2022-02-21

**Authors:** Sintayehu Assefa Endaylalu, Wei-Hsin Tien

**Affiliations:** Department of Mechanical Engineering, National Taiwan Science and Technology University, No. 43, Section 4, Keelung Road, Da’an District, Taipei City 106, Taiwan; d10703811@mail.ntust.edu.tw

**Keywords:** acoustic streaming, micromixer, acoustofluidics, microfluidics, computational fluid dynamics

## Abstract

In this study, the mixing performance in a Y-junction microchannel with acoustic streaming was investigated through numerical simulation. The acoustic streaming is created by inducing triangular structures at the junction and sidewalls regions. The numerical model utilizes Navier–Stokes equations in conjunction with the convection–diffusion equations. The parameters investigated were inlet velocities ranging from 4.46 to 55.6 µm/s, triangular structure’s vertex angles ranging from 22° to 90° oscillation amplitude ranging from 3 to 6 µm, and an oscillation frequency set to 13 kHz. The results show that at the junction region, a pair of counter-rotating streaming vortices were formed, and unsymmetrical or one-sided vortices were formed when additional triangles were added along the sidewalls. These streaming flows significantly increase the vorticity compared with the case without the acoustic stream. Mixing performances were found to have improved with the generation of the acoustic stream. The mixing performance was evaluated at various inlet velocities, the vertex angles of the triangular structure, and oscillation amplitudes. The numerical results show that adding the triangular structure at the junction region considerably improved the mixing efficiency due to the generation of acoustic streaming, and further improvements can be achieved at lower inlet velocity, sharper vertex angle, and higher oscillation amplitude. Integrating with more triangular structures at the sidewall regions also improves the mixing performance within the laminar flow regime in the Y-microchannel. At Y = 2.30 mm, oscillation amplitude of 6 µm, and flow inlet velocity of 55.6 µm/s, with all three triangles integrated and the triangles’ vertex angles fixed to $30^{{}^{\circ}}$, the mixing index can achieve the best results of 0.9981, which is better than 0.8355 in the case of using only the triangle at the junction, and 0.6642 in the case without acoustic streaming. This is equal to an improvement of 50.27% in the case of using both the junction and the two sidewall triangles, and 25.79% in the case of simply using a junction triangle.

## 1. Introduction

Fluid mixing is a phenomenon process in microfluidics that involves the combination of various materials. Homogeneous and rapid mixing in microscales are critical for various applications such as drug delivery, chemical process industries, biomedical diagnostics, etc. The main mixing mechanism is governed by molecular diffusion. Micromixers are thus integrated into microfluidic devices to achieve quick mixing by facilitating mass transfer between the flow streams. They can be categorized as either active or passive depending on the amount of energy consumed [1]. Passive micromixers are easy to set up and integrate with other microfluidic components. However, this type of micromixer has less performance compared with active micromixers. The different geometry of the microchannel section is modified in passive micromixers to induce flow disturbance as well as pressure difference, which then promotes mixing quality. The most common geometrical features used as passive micromixers are swirl inducing [2], obstructions [3], the placement of obstacles [4], etc. Active micromixers perform mixing to enhance quality and speed by utilizing external power sources such as acoustic field [5], pressure-driven [6], electric field [7], and magnetic-field [8]. Active micromixers can be activated based on the user’s needs and provide controllable mixing with electrical voltage, pressure gradient, and integrated elements. In general, active micromixers improve mixing by magnetically, mechanically, electrically, and acoustically swirling the fluid streams.

Acoustofluidics is a combination of acoustics and microfluidics. It has many application areas such as microfluidic pump [9], bimodal signal amplification [10], mixing and migration of micro-size particles [11], etc. Acoustic streaming occurs as a result of the time-averaged nonlinear dynamics of the solid–fluid interaction in a viscous flow [12,13,14]. It is induced by the second order time-averaged streaming fluctuating components in any fluid flow. Ovchinnikov et al. studied numerically the flow pattern and scaling around a single sharp edge and found that it generated a high Reynold body force when compared with its non-sharp counterparts [15]. Similarly, Doinikov et al. recently reported that sharp edge structures were used to generate acoustic streaming in a circular microfluidic device of the fluid domain [16]. Both studies indicated that the sharp edge of the solid body is used as the origin of acoustic streaming to oscillate the flow. The acoustic streaming production is strong enough near the apex with a small vertex angle and is also directly related to the applied amplitude [17]. Its generation is also affected by the fluid’s Reynold number flow characteristics. Generally, the induced streaming flow pattern is stronger at low Reynold number flows than at high Reynold number flows [18]. The acoustic stream is produced more efficiently when the radius of the curvature of the apex is small in comparison with the thickness of the viscous boundary layer [19]. Acoustic streaming has been recognized as an important and non-invasive solution for different applications such as mixing [20,21], heat transfer [22], synthesis of organic nanoparticles [23], and particle patterning [24]. Nama et al. numerically investigated the performance of micromixing by inducing acoustic oscillated sharp edges in a sidewalls rectangular computational domain [25]. In general, active and passive mixing in Y-microchannels has previously been performed such as acoustically induced bubbles [26,27], rotating magnetic fields [28], placement of obstacles in the microchannels [29,30], electronically driven flow [31], acoustic streaming around the sharp triangular structure in the sidewall directions [20], adding split and recombination features [32], side lateral obstructions [33], and symmetrical cylindrical grooves [34].

The Y-microchannel is a micromixer configuration used in microfluidics devices. Different geometries such as a circle, square, rectangle, various inlet and outlet section regions, and channel angles are developed, but its configuration is mainly defined by two inlets and one outlet, with the single outlet acting as a micromixer at various angles [35]. The mixing of two or more fluids is a critical process inside the Y-microchannel. Since the fluid flow inside the small dimension of the Y-microchannel is mostly laminar, it is dependent on its outlet effective length and diffusion disturbance to achieve the maximum optimum species mixture concentration at the common microchannel outlet region. Thus, fluid flow pattern disturbance is required before passing through the junction region to minimize the effective outlet length and also reduce the microchannel manufacturing costs.

The main objective of this study is to investigate numerically the mixing performance of acoustic streaming in a Y-microchannel by inducing a triangular structure starting from the microchannel junction region. It can achieve quick mixing and improve homogeneity within the microchannel mixing section by disrupting the flow pattern starting from the junction region and generating a vortex streaming flow. The paper is organized as follows: methods are presented in Section 2, results and discussion are presented in Section 3, and Section 4 concludes with the important findings and future work.

## 2. Numerical Methods

### 2.1. Microchannel Geometry and Numerical Scheme

Figure 1 demonstrates the CAD design of the microchannel used in the current study. The Y-junction microchannel segments are separated by a 120° angle. Figure 1a illustrates the dimensions and configuration of the common standard Y-junction microchannel. The new microchannel configuration includes fillet dimensions to reduce the microchannel edge effect and enlarge the junction region. The triangle’s apex inside the microchannel is rounded with 0.30 µm, which is below the viscous boundary layer thickness. Similarly, the triangle has a characteristic height, h, around the junctions of 0.35 mm, 0.25 mm, and 0.17 mm at the microchannel outlet sidewalls, which is equal to the distance between the triangle’s tip and the beginning of its base. Figure 1b,c show the new microchannel designs for numerical modeling. It was designed with an induced single triangular structure only at the junction region and the addition of two more triangles to the microchannel sidewalls at the outlet direction. As shown in Figure 1c, the first sidewall triangle induced 0.40 mm away from the apex of the junction triangle, while the second triangle induced in the other side at a distance of S away from the first sidewall triangle’s apex. The triangle’s apex is located between the two edges of the outlet microchannel, as shown in Figure 1b, but below 0.10 mm in the case of the 0.25 mm triangle height, h, as shown in Figure 1c. The base edges of the triangular structure inside the microchannel are rounded with 0.10 mm and 0.02 mm, as shown in Figure 1b,c.

The Y-junction microchannel numerical model is in the coordinate rectangular region ranging in *x*-axis (0.4398, 0.2598) to (2.4255, 0.2598) and in *y*-axis (0.5898 and 2.2755, 0) to (1.1326 and 1.7326, 2.3484) as shown in Figure 2. The origin of the coordinate system is set to be at the lower-left corner of Figure 2. Throughout this study, the origin is fixed at this point and the coordinate system is unified in order to express all the data and results. The Y-shaped microchannel has two inlets and one outlet microchannel for mixing purpose. The lower and inlet side microchannel edges of the 2D modeling are 30° and 60° from the *x*-axis, respectively. Similarly, the upper and inlet side microchannel edges are 60° and 30° from the *y*-axis, respectively.

The numerical model scheme is used to solve the two-dimensional water domain of the Y-microchannel cross section, which has two inlets and one outlet segment. The material properties used in the numerical study are listed in Table 1. Figure 2 shows the geometry of the numerical model discretization of the new Y-junction microchannel. The numerical model was implemented and the governing equation solved using the finite element software COMSOL Multiphysics [36]. Three sets of governing equations are used to obtain the results in this study. Based on the temporal and spatial scales, the acoustic velocity field is first calculated by using the thermoviscous acoustics module in frequency domain. The streaming flow velocity field is then calculated by applying the laminar flow physics module. Lastly, the transport of diluted species module is used to solve the convective transport of a solute species from one inlet side to the outlet’s common mixing region in the Y-junction microchannel. Table 2 shows the numerical study input parameters.

### 2.2. Governing Equations and Boundary Conditions 

Acoustic streaming fundamental governing equations have been presented in various studies [38,39]. The perturbation theory is used to solve microchannel flow problems in the context of “weak disturbance”. It is an effective tool for reducing the Navier–Stokes equation, which includes the nonlinear terms that couple the acoustic and streaming velocity fields. The general governing equations are continuity, momentum, and energy equations. The numerical model was performed in three steps: (i) solving the wave equation to compute the acoustic velocity fields, (ii) computing the streaming flow, and (iii) solving the flux of the concentration profile in the microchannel. 

In the thermoviscous acoustics module, governing equations are obtained from the linearized Navier–Stokes equations, which are used to solve the continuity and momentum in the microchannel system. All governing equations fields and sources are assumed to be harmonic with $e^{i\omega t}$. Therefore, the acoustic velocity field was determined in the following ways:(1)$$i\omega \rho_{a}+\nabla .\left(\rho_{o}v_{a}\right)=0$$
(2)$$i\omega \rho_{o}v_{a}=\nabla .\left[-PI+\mu \left(\nabla v_{a}+{\left(\nabla v_{a}\right)}^{T}\right)-\left(\frac{2}{3}\mu -\mu_{B}\right)(\nabla .v_{a})I\right]$$
where ω is the frequency of actuation,$\;v_{a}$ is the acoustic velocity field, and $\rho_{a}$ is the density at temperature $T_{a}$:(3)$$\rho_{a}=\rho_{o}\left(\kappa_{T}p_{a}-\alpha T_{a}\right)$$
(4)$$\kappa_{T}=\frac{1}{\rho }{\left(\frac{\partial \rho }{\partial p}\right)}_{T}=\frac{1}{\rho_{o}c_{o}{}^{2}}$$

$c_{o}$ is the speed of sound in fluid, $\kappa_{T}$ is the isothermal compressibility coefficient, and  α is the coefficient of thermal expansion.
(5)$$\alpha =\frac{1}{\rho }{\left(\frac{\partial \rho }{\partial T}\right)}_{P}=\frac{1}{c}\sqrt{\frac{C_{p}\left(\gamma -1\right)}{T}}$$

*µ* and $\mu_{b}$ are the viscous and bulk dynamic viscosities, respectively. I is the identity matrix. The walls of the microchannel are solid surfaces with no-slip and isothermal boundary conditions. The thermoviscous acoustics length scale is defined by viscous and thermal boundary layer thicknesses, but due to the low oscillation frequency in kHz level, it is sufficient to describe it with only the viscous boundary layer thickness. Therefore, the viscous penetration depth (viscous boundary layer thickness) is given by: (6)$$\delta =\delta_{v}=\sqrt{\frac{2\mu }{\omega \rho }}=\sqrt{\frac{\mu }{\pi f\rho }}=\sqrt{\frac{\nu }{\pi f}}$$

According to Equation (6), the thickness of the viscous boundary layer decreases as the frequency, *f*, of the oscillation increases, but it can also increase as the frequency, *f*, decreases.

The ratio of inertia to viscous forces, i.e., the Reynold number $\left(R_{e}=\left(\rho vD_{h}\right)/\mu \right)$, is small, indicating that viscous forces dominate and damp out all disturbance in the microchannel. Similarly, acoustic streaming at the sharp edge has a dominant body force over other driving forces in the microchannel. As a result, the fluid flow inside the microchannel is in a single phase and operates in a laminar flow regime. Likewise, the acoustic wavelength (λ=c/f) is much larger than the width dimension of the microchannel, where c is the speed of sound in water. The dimensionless number of a fluid flow Mach number (${\mathrm{Ma}=v}_{a}/c\ll 1)$ and the temperature variation in the microchannel are very small, and the density is also nearly constant. As a result, the laminar fluid motion is assumed to be incompressible, and the governing equations of the continuity and Navier–Stokes momentum equations are as follows in Equations (7) and (8), respectively:(7)ρ∇.v=0
(8)$$\frac{\partial v}{\partial t}+\left(v.\nabla \right)v=\frac{1}{\rho }\left(\nabla .(-pI+\mu \left(\nabla v+\left(\nabla v{)}^{T}\right)\right)+F\right)$$
where $F=\langle \rho_{a}\frac{\partial v}{\partial t}\rangle +\rho_{o}\langle \left(v.\nabla \right)v\rangle$ is the volume force (N/$m^{3})$ and v is the streaming velocity field.

As stated previously, the microchannel has a solid wall and is stationary. The wall boundary condition is therefore no-slip and zero velocity is assumed. Similarly, the fluid flow at both inlets is considered as fully developed and uniform, so the inflow boundary condition is assigned by the average velocity. The pressure boundary condition is equal to zero at inlets and absolute pressure $p_{o}$ at the outlet.

The transport of fluorescein sodium salt species in the microchannel via convection-diffusion is solved using Equation (9):(9)$$\frac{\partial c_{i}}{\partial t}+\nabla .J_{i}+v.\nabla c_{i}=R_{i}$$
where $c_{i}$ is the concentration of the species ($\mathrm{mol}/m^{3})$, $R_{i}$ is a reaction rate expression for the species ($\mathrm{mol}/\left(m^{3}\cdot s\right))$, and v is the streaming fluid averaged velocity vector (m/s). $J_{i}$ denotes the mass flux factors that define the diffusive flux vector and are solved using Equation (10): (10)$$J_{i}=-D_{i}\nabla c_{i}$$
where $D_{i}$ is the diffusion coefficient of fluorescein sodium salt species ($m^{2}/s$). 

The initial boundary condition concentration $c_{o}$ of the material species was set to 1 mol/$m^{3}$ in one inlet direction and 0 in the other inlet side. 

The mixing index or degree of mixedness is investigated based on the statistical method. It is calculated at any cross section of the microchannel width from the standard deviation of the fluorescein sodium salt species concentration using Equation (11) [40]:(11)$$M=1-\sqrt{\frac{\sigma^{2}}{\sigma^{2}{}_{max}}}$$
(12)$$\sigma =\sqrt{\frac{1}{n}\sum_{i=1}^{n}\left(c_{i}-\bar{c}\right)}$$
(13)$$\sigma_{\max}=\sqrt{\bar{c}\left(1-\bar{c}\right)}$$
where σ is the standard deviation of the concentration species $c_{i}$ in each point on the width of the cross section, $\bar{c}$ is the mean of the concentration $c_{i}$ on the width cross section, and $\sigma_{\max}$ is the maximum standard deviation of the species concentration at the specified width cross section of the microchannel. 

### 2.3. Mesh Independence Test

The mesh independence test is required in the simulation to find the optimal mesh. An optimal mesh is one in which the computational time is minimized while the accuracy of the result is maintained. The independence test was evaluated by considering the acoustic velocity fields and streaming velocity magnitude at various mesh element sizes. The computational mesh is created by combining a maximum element size length $d_{\mathrm{mesh}}$ at the domain boundaries and bulk of the fluid. The bulk water domain region mesh size $d_{\mathrm{mesh},\mathrm{dk}}$ is kept constant with 12δ, where δ is the thickness of viscous boundary layer thickness. Then, the mesh independent test was verified by different boundary element sizes, $d_{\mathrm{mesh},\mathrm{db}}$. Moreover, the smallest mesh element size $d_{\mathrm{mesh},\mathrm{db}}=0.25\delta$ is taken as a reference computational mesh element size and evaluated the mesh independent test in terms of relative convergence parameter C(g) using Equation (14) as follows [41]:(14)$$C\left(g\right)=\sqrt{\frac{\int {\left(g-g_{\left(ref\right)}\right)}^{2}dxdy}{\int {\left(g_{\left(ref\right)}\right)}^{2}dxdy}}$$
where *g(ref)* is the reference value at the smallest mesh element size.

The evaluation was performed at the junction tip edge triangles up to 10 µm from the numerical model, as shown in Figure 2a. Figure 3 shows the relative convergence parameter C vs. $\delta /d_{\mathrm{mesh},\mathrm{db}}$, demonstrating that the value of the parameter C(g) is very small in all ranges of mesh element size. As a result, $\delta /d_{\mathrm{mesh},\mathrm{db}}=1.33$ is chosen for all succeeding study cases. The velocity fields $v_{1x}$ and $v_{1y}$ are for acoustic velocity components, and $v_{2}$ is the velocity magnitude for streaming flow.

When the size of the boundary mesh element is reduced to a minimum, the relative convergence parameter approaches zero as shown in Figure 3.

## 3. Results and Discussion 

### 3.1. Normal Y-Junction Microchannel

Figure 4 illustrates the flow vector plot, velocity magnitude, and species concentration distribution in the original Y-junction microchannel without the triangle structure and acoustic field. The fluid flow from the inlet side of the microchannel followed its path to the microchannel outlet as demonstrated in Figure 4a. Because of the layered stream formed between the two side fluid streams, concentrated species were difficult to diffuse and convect to the other sides of the microchannel outlet region as shown in Figure 4b.

### 3.2. Acoustic Streaming Generation

The Y-junction microchannel has an induced triangle edged in the junction and also integrated with the sidewall triangle’s edges, which are positioned in an oscillation fluid to create a vortex stream in both clockwise and anti-clockwise rotation. The generated acoustic field and streaming flow problems are addressed using perturbation theory. The viscous boundary thickness was much less than the height dimensions of the microchannel-induced triangle’s sharp edge structure. The triangle’s sharp edge is rounded to avoid a numerical solution discontinuity purpose. However, its radius curvature of the round is sufficiently sharper in comparison with the viscous boundary layer thickness dimension, i.e., r≪δ. This implied that the length scale was enough to create acoustic streaming flow.

Figure 5 shows the quiver plot and velocity magnitudes after inducing triangular structures in different positions. The streaming flow originated from the two inlet sides, whereas acoustic streaming flows began from the triangle’s sharp edges. Acoustic stream line patterns followed a revolving circular structure and produced strong vorticity on both sides of the induced triangular structure. Figure 5a shows the streaming flow and revolving caused by inducing only the triangular structure around the junction region. Streaming velocity has a high magnitude of streaming velocity around the sharp tip edges, but produces two strong counter-rotating vortices on both sides of the triangle regions. The strength of the vortices expressed interims of vorticity. It was extracted and plotted on both the left and right sides at equal distances from the tip edge, then compared in different junction triangle vertex angles. 

The side walls of the isosceles triangle vibrated highly in the study conducted in two-dimensional X- and Y-directions. The results indicated that two sides streaming layers were created starting from the tip sharp edge during the induction of one triangular structure at the junction region, as shown in Figure 5a. When the sidewalls triangles were induced, the layer’s curve length size was minimized and rotated in a similar manner into two circular counter vortices, as shown in Figure 5b–f. First and foremost, the effect of the placement of the gap, S, between the two sidewall triangles was considered and investigated. When the spacing between the sidewall triangles was reduced to 0.15 mm, it produced a short curved revolving streaming from the first right sidewall triangle’s apex to the second left sidewall triangle edge, as shown in Figure 5b, but when the space, S, was increased to 0.90 mm, the stream pattern became less curved and the second left sidewall triangle created two unsymmetrical streaming vortices, as shown in Figure 5f.

Figure 6b shows symmetrical vortices and high vorticity on both sides of the junction-induced triangle, as plotted by cutting plane lines X = 1.283 mm, X = 1.583 mm, and Y = 0.485 to 2.30 mm. There are clockwise and anti-clockwise vortices on both sides of the triangular structure. Therefore, there is a higher tip velocity maximum as well as vorticity at the very sharp-edged tip of 22° than 60°. It has a maximum velocity of around 5 mm/s on a 30° triangle tip sharp edge angle, as shown in Figure 5. Figure 5c also shows the vorticity generation at a specified geometry coordinate region of both sides of the junction triangle in different oscillation frequencies from 3 to 13 kHz. The 13 kHz oscillation frequency has better vorticity generation performance. Therefore, this frequency is selected for the input parameter to evaluate all numerical study work.

### 3.3. Mixing Performance 

The acoustic streaming produced increased the disturbance of the concentrated species beginning at the junction of the Y-microchannel region as shown in Figure 7.

The diffusional movement of concentrated species followed the streaming pattern and more disturbances originated from the junction triangle’s apex. Due to the high streaming velocity around the sharp edge compared with other regions, it created a high-streaming curved layer. Therefore, with only a sharp edge at the Y-junction, mixing was incomplete and did not to fully achieve the desired objective. Another microchannel design and the number of integrated triangles was used to improve the mixing performance of the microchannel. Figure 7b–f shows the visualization of successive mixing performance after integrating two sidewall triangles and shortening the two strong layers created in the junction triangle’s apex.

The effect of streaming velocity and vortices strength on the concentration species distribution in the microchannel outlet region was evaluated using only the junction region induced triangular structure, as shown in Figure 7a. The mixing performance was evaluated using the concentration flux in comparison with the expected optimum concentration of the species across the width of the outlet microchannel and also the dimensionless parameter mixing index along the flow direction in the measurement region as shown in Figure 8. The width of the microchannel outlet ranges of from X=1.1326 to 1.7326 mm along the horizontal axis. 

The performance of acoustic streaming on mixing is affected by variables such as inlet velocity, oscillation amplitude, and the vertex angles. Considering the evaluation mechanism of mixing effectiveness is a concentration profile in the microchannel perpendicular to the flow direction in comparison with the expected optimum species concentration. At the end of the mixing process, the expected optimum concentration species is 0.50 mol/m^3^ inside the outlet microchannel. Figure 9 illustrates the species concentration profile at Y = 2.30 mm perpendicular to the flow direction with various parameters. As shown in Figure 9b, the concentration profile resulted in less than the expected optimum concentration in all cases due to the low inlet velocity as well as the junction and sidewalls with the same vertex angle. However, it was improved at high inlet velocity, as shown in Figure 9c. Similarly, it improved sufficiently at low inlet velocity when the junction triangle’s vertex angle was more than the sidewall triangle tip angles, as displayed in Figure 9d.

Acoustic streaming produces more body force around sharp edges than other non-microchannel parts, which is expected to result in a 3D flow phenomenon. However, the mixing process takes place in small microchannel dimensions and follows the bulk movement of fluids. The Y-microchannel must always have an inlet and an outlet with a continuous flow process. As a result, the mixing performance is always evaluated far from the high disturbance and around the outlet cross section. Therefore, it is one of the primary reasons to focus on mixing performance evaluation in 2D. Figure 10 illustrates the concentration profile when the sidewall triangle’s vertex angle is reduced from $\alpha_{2}$ = 30° to 50°. The results show that the sidewall triangles at a distance of S = 0.15 mm and 0.30 mm delivered close to the expected optimum concentration flux profile compared with other dimensions, as shown in Figure 10a. Similarly, low inlet velocity has better output performance than high inlet velocity, as displayed in Figure 10b, but has worse output performance at lower oscillation amplitude, as seen in Figure 10c.

The mixing index M is determined using Equation (11) by a concentration of species taken in the measurement region between y = 0.75 to 2.30 mm in all cases, as shown in Figure 8. It is increased along the flow direction and its acoustic streaming performance is also more effective at low inlet velocity, as indicated in Figure 11a,d,e. Figure 11a illustrates a mixing index profile ranging from 0.631 to 0.88 after inducing a single triangle at the junction region with an inlet velocity of 4.46 m/s, however, when the inlet velocity increased to 55.6 m/s, the performance, M, profile changed from 0.45 to 0.84.

Mixing performance is improved with the same inlet after sidewall triangles are induced and integrated. The distance, S, between the sidewall triangles was investigated with different values, as shown in Figure 5, Figure 7, Figure 9b,c and Figure 10a. The distance, S, was better at 0.30 mm based on the streaming vortices profile shown in Figure 5 and the expected optimum concentration profile shown in Figure 9b. Moreover, the dimensionless mixing index, M, was investigated for all values of S, as shown in Figure 11b,c. However, the mixing index M value did not result in an exaggerated result at the microchannel outlet in all S value cases except around the sidewall triangles regions. In general, the sharp edge design of three integrated triangles performed better than the sharp edge design of a single junction triangle. As shown in Figure 11f, the sharper edge of the sidewall provided a better mixing index than the junction triangle’s apex, but its concentration profile was lower than the expected optimum concentration value at Y = 2.30 mm plane cutting line. As a result, to satisfy those conditions at low inlet velocity, the junction triangle’s vertex angle should be sharper than the sidewall triangles’.

In general, when compared with a standard Y-junction microchannel, the generated acoustic streaming disturbance has a significant effect on fluid disturbance and improved mixing performance. It performed well in both the concentration profile with a reference to the expected optimum concentration of 0.50 mol/m^3^, as indicated in Figure 12, and the mixing index M, as shown in Figure 13 below. 

The concentration of species profile varied between 0.7129 and 0.2871 mol/m^3^ in the normal Y- junction microchannel cases, whereas with one triangle at the junction, the species concentration profile ranged between 0.6008 and 0.3989 mol/m^3^, but using three triangles improved its range between 0.5015 and 0.4991 mol/m^3^.

Similarly, at Y = 2.30 mm, the mixing index M performance of a normal Y-junction microchannel, with only one triangle at the junction, and an integrated three triangles is 0.6642, 0.8355, and 0.9981, respectively.

## 4. Conclusions

In this study, a new configuration of acoustofluidics was proposed to improve the mixing performances of a Y-junction micromixer. By introducing acoustic streaming with triangular structures at the junction and sidewall regions, the acoustic streaming vortices created by the structure at the junction can improve the mixing efficiency, and the streaming vortices generated by this triangular structure at the junction region can be further integrated with two additional ones at the channel sidewalls to produce successive vortices in the mixing channel to elongate the mixing enhancement. Through numerical simulations, the strength of the vortices is evaluated in terms of the Z-vorticity magnitude, whereas the mixing effectiveness is measured by the concentration profile across the width of the outlet microchannel section and the dimensionless mixing index, M, in the measurement region X = 1.1326 to 1.7326 mm and Y = 0.75 to 2.30 mm along the flow direction. Conclusions can be drawn from the obtained results:

First, introducing the acoustic streaming considerably increased the vorticity in the flow field, and therefore increased the mixing index. In the present study, the Y-junction microchannel without acoustic streaming had a mixing index of 0.6642. By introducing the acoustic streaming with a single triangular structure at the junction region, the mixing performance improved to 0.8355. Three triangular structures at the junction and sidewall regions further improved the result to 0.9981 at Y = 2.30 mm with an inlet velocity of 55.6 µm/s. This corresponds to an increase of 25.79% and 50.27% in the case without acoustic streaming, respectively.

Second, the strength of the streaming vortices and mixing performance are influenced by the parameters such as inlet velocity and the vertex angles of the triangular structure. Higher inlet velocities result in slightly worse mixing performance. In the present study, inlet velocities were tested from 4.46 μm/s to 55.6 μm/s. The mixing index, M, gradually decreased from 0.880 at 4.46 μm/s inlet to 0.8355 at inlet velocity 55.6 μm/s. The same trends can be observed for the cases with three triangular structures, decreasing from 0.9989 to 0.9981 at low and high inlet velocities, respectively. Sharper vertex angles produced higher vorticities and more uniform species distributions in the microchannel. To achieve even better performances in terms of both concentration profile and mixing index, the junction triangle’s apex should be sharper than the sidewall triangles’. The sharper case, $\alpha_{2}{=30}^{{}^{\circ}}$, has a mixing index of 0.9989, while the case of $\alpha_{2}{=90}^{{}^{\circ}}$ only has a mixing index of 0.9729. Further optimization of the geometrical details might be achieved by adopting methods such as topology optimization, if certain design restrictions or requirements can be specified to form the objective function. However, the results might be sensitive around the apexes of the triangular structures due to the higher curvature.

Finally, the Z-vorticity calculated from the velocity fields correlated with the mixing performance well. The improvement of performances, namely, the species concentration profile with respect to optimum concentration and mixing index, M, due to acoustic streaming, can be explained by the increase in the vorticity magnitude in the vorticity map. This correlation can be applied to experimental studies to evaluate the mixing performance using velocimetry techniques such as micro-particle image velocimetry (μ−PIV). More ongoing works are currently conducted to investigate the flow patterns experimentally and take the three-dimensional boundary effects into account, such as the channel height and vertical triangular structure.

## Figures and Tables

**Figure 1 micromachines-13-00338-f001:**
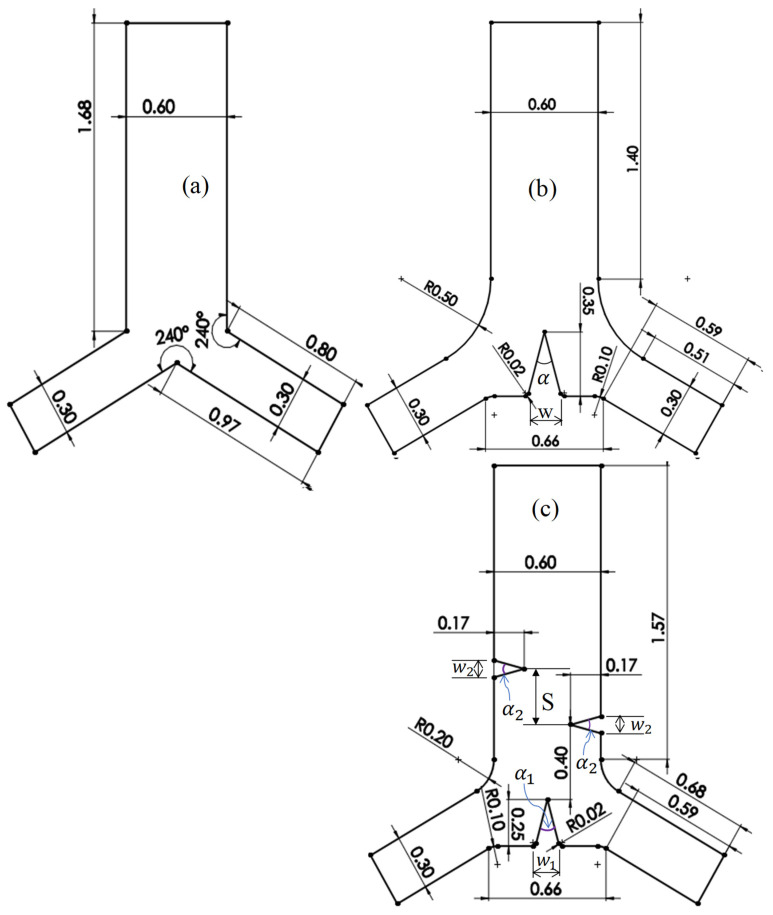
Designs of Y-microchannel configurations of the present study: (**a**) normal Y-channel without triangle structures, (**b**) triangular structure for acoustic streaming at the junction only, and (**c**) triangular structures in the junction and the channel sidewalls. All dimensions are marked in mm. In (**b**), where α is the triangle’s vertex angle, w is the width of the triangle; in (**c**), where S is the sidewall triangle’s spacing, $\alpha_{1}$ and $\alpha_{2}$ are the vertex angles of the triangles at the junction and the sidewalls, respectively. ${𝑤}_{1}$ and ${𝑤}_{2}$ are the widths of the junction and sidewall triangles, respectively.

**Figure 2 micromachines-13-00338-f002:**
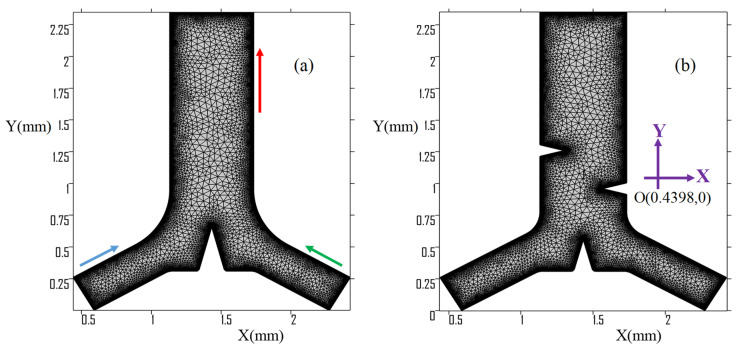
Geometrical mesh; (**a**) with only one junction triangle; (**b**) with three triangles.

**Figure 3 micromachines-13-00338-f003:**
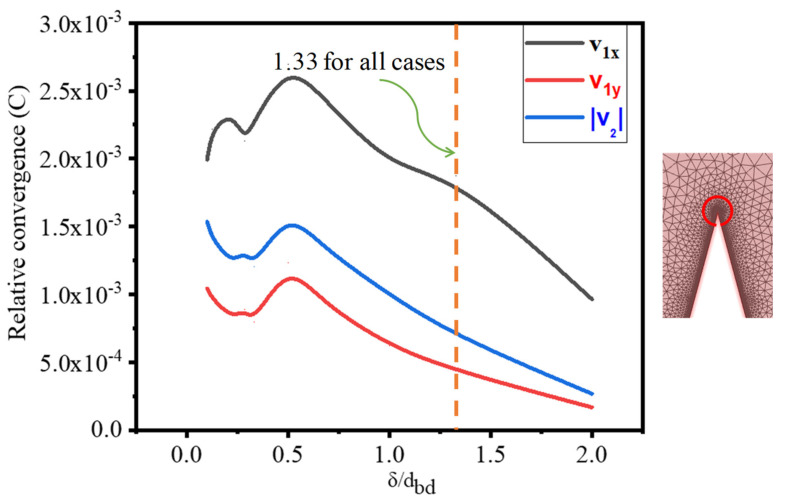
Mesh independence test.

**Figure 4 micromachines-13-00338-f004:**
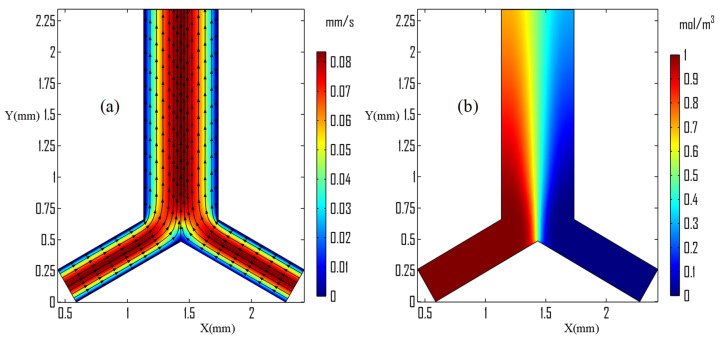
Visualization in standard Y-junction microchannel at 55.6 μm/s; (**a**) velocity vector and magnitude; and (**b**) species concentration distribution.

**Figure 5 micromachines-13-00338-f005:**
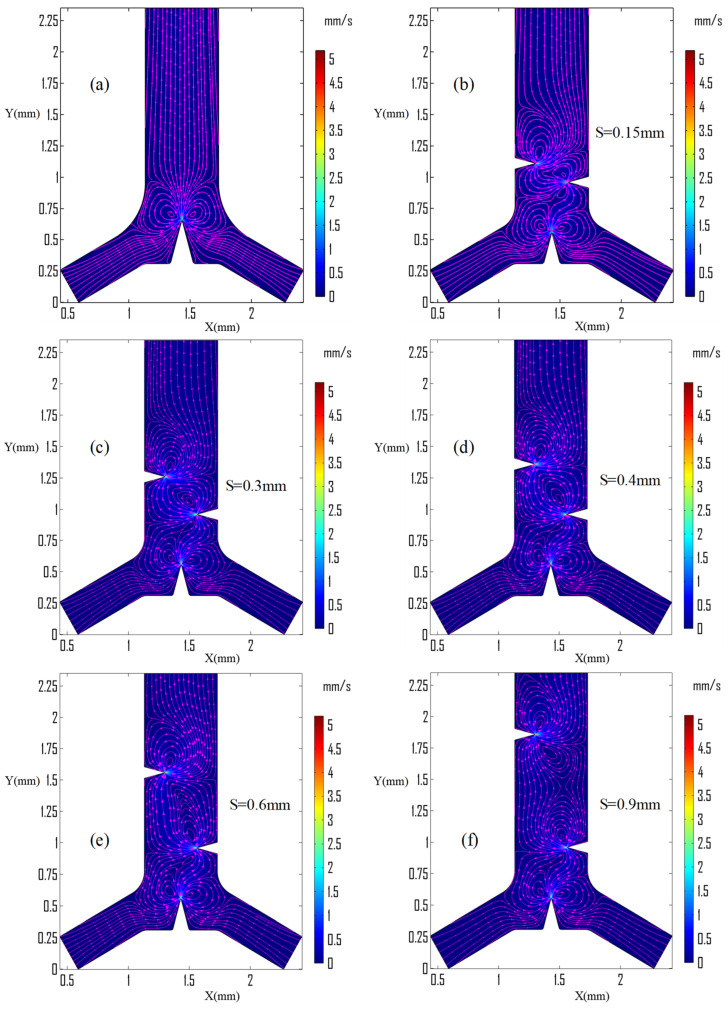
Acoustic streaming generation at 13 kHz, 6 μm amplitude, 4.46 μm/s inlet velocities on ${\alpha =30}^{{}^{\circ}}{,\;\alpha }_{1}{=30}^{{}^{\circ}}{,\;\alpha }_{2\;}{=30}^{{}^{\circ}}$; (**a**) only inducing the triangular structure on junction region; and (**b**–**f**) at different sidewalls triangles spacing.

**Figure 6 micromachines-13-00338-f006:**
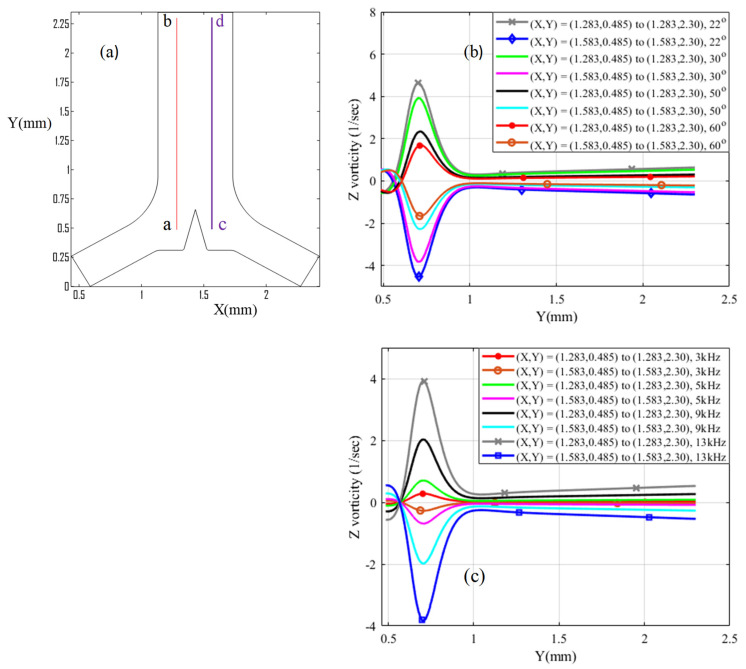
Vorticity comparison: (**a**) cutting plane line $\bar{ab}$ and $\bar{cd}$ position in (X,Y) coordinate system; (**b**) Z vorticity comparison using 13 kHz, 6 μm, and 4.46 μm/s in different α of junction triangle’s vertex angles; and (**c**) Z vorticity using 6 μm and 4.46 μm/s on α = 30° junction triangle vertex angle in different oscillation frequency.

**Figure 7 micromachines-13-00338-f007:**
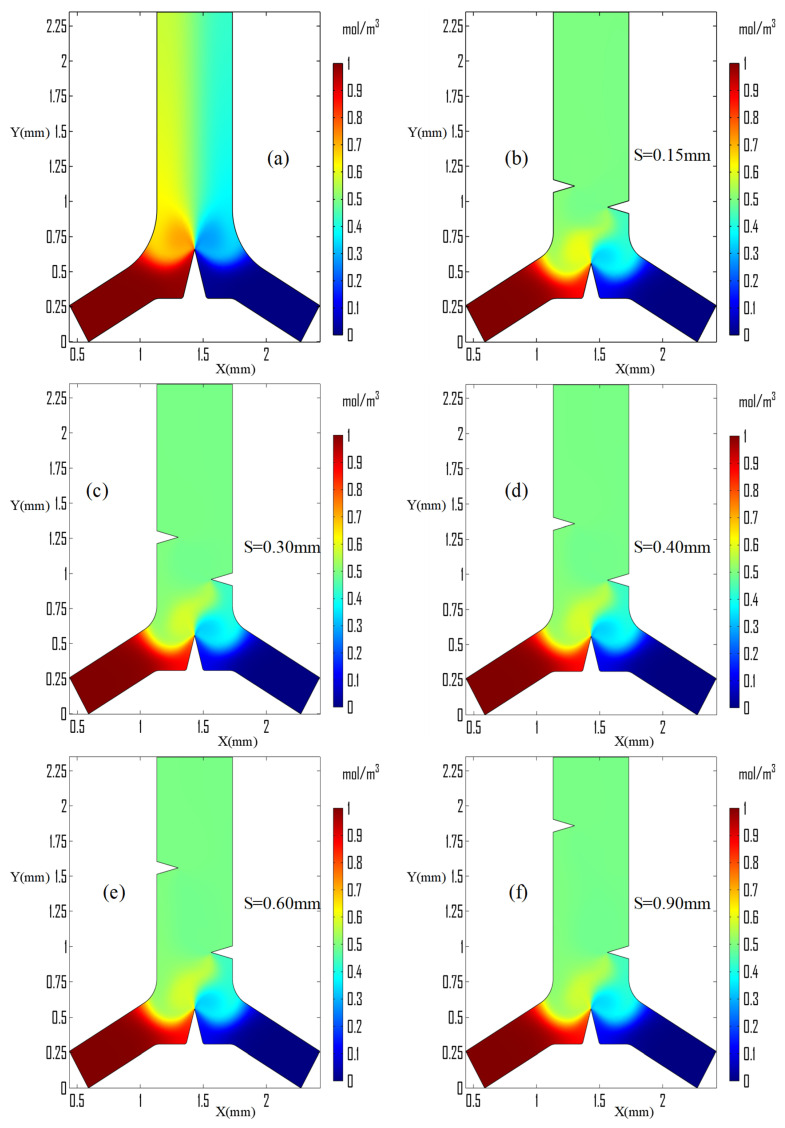
Mixing performance visualization at 13 kHz, 6 μm amplitude, 4.46 μm/s inlet velocity on ${\alpha =30}^{{}^{\circ}}{,\;\alpha }_{1}{=30}^{{}^{\circ}}{,\;\alpha }_{2\;}{=30}^{{}^{\circ}}$; (**a**) only inducing the triangular structure on junction region; and (**b**–**f**) at different sidewalls triangles spacing.

**Figure 8 micromachines-13-00338-f008:**
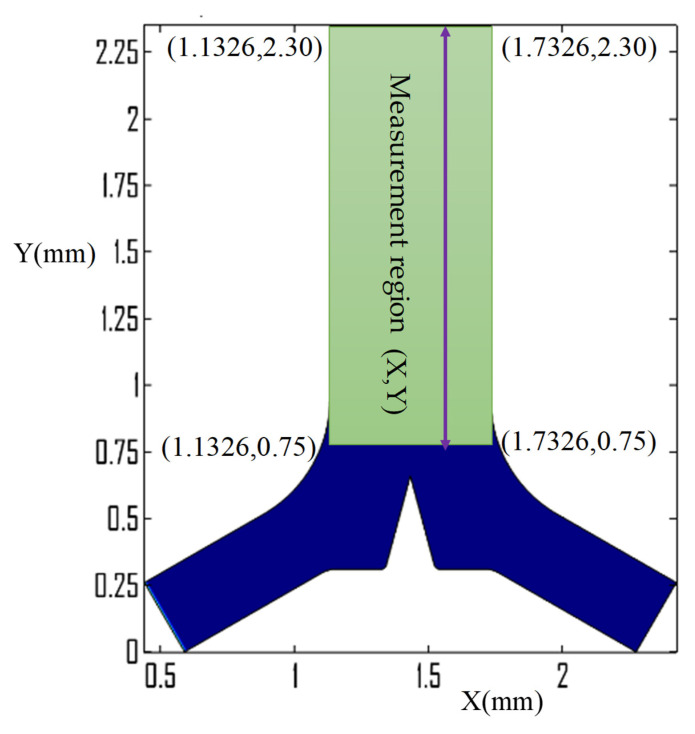
Mixing evaluation region in the Y-microchannel.

**Figure 9 micromachines-13-00338-f009:**
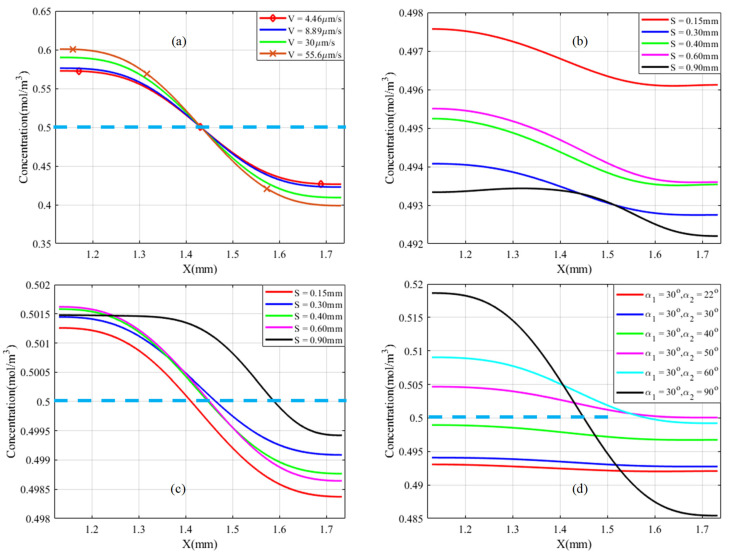
Species concentration profile comparison with 13 kHz, 6 μm oscillation amplitude at Y = 2.30 mm; (**a**) only junction ${\alpha =30}^{{}^{\circ}}$; (**b**) at $\alpha_{1}{=30}^{{}^{\circ}},\alpha_{2\;}{=30}^{{}^{\circ}}$, at 4.46 μm/s with different S; (**c**) at $\alpha_{1\;}{=30}^{{}^{\circ}},\alpha_{2}{=30}^{{}^{\circ}}$, at 55.6 μm/s inlet velocity with different S; and (**d**) at 4.46 μm/s inlet velocity and S = 0.30 mm.

**Figure 10 micromachines-13-00338-f010:**
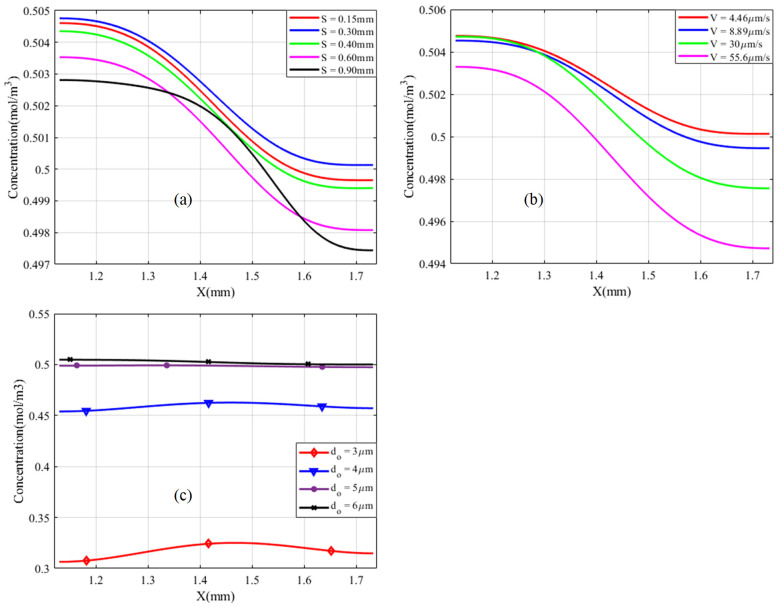
Concentration profile comparison with 13 kHz at Y = 2.30 mm, and $\alpha_{1}{=30}^{{}^{\circ}},\alpha_{2\;}{=50}^{{}^{\circ}}$; (**a**) at 4.46 μm/s with different S; (**b**) $\alpha_{1\;}{=30}^{{}^{\circ}},\alpha_{2}{=50}^{{}^{\circ}}$, S = 0.30 mm, and at different inlet velocity; and (**c**) $\alpha_{1}{=30}^{{}^{\circ}},\alpha_{2}{=50}^{{}^{\circ}}$, S = 0.30 mm, 4.46 μm/s inlet velocity, and at different oscillation amplitudes.

**Figure 11 micromachines-13-00338-f011:**
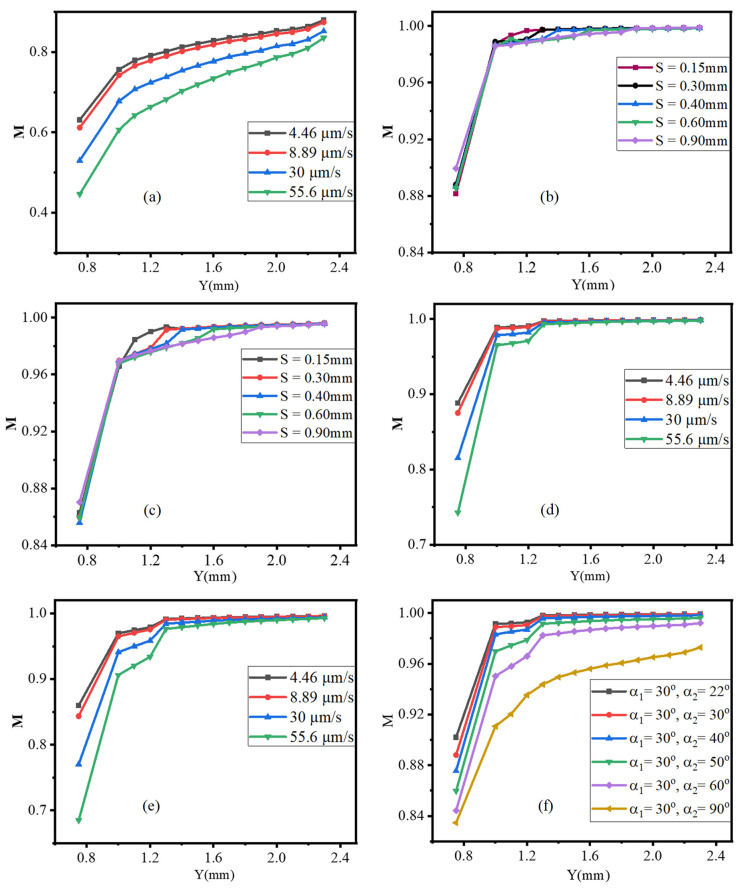
Mixing index with 13 kHz, 6 μm, (**a**) only induced triangular structure at junction with ${\alpha =30}^{{}^{\circ}}$; (**b**) at $\alpha_{1}{=30}^{{}^{\circ}}{,\alpha }_{2\;}{=30}^{{}^{\circ}}$, at 4.46 μm/s with different sidewalls triangles spacing; (**c**) at $\alpha_{1}{=30}^{{}^{\circ}},\alpha_{2}{=50}^{{}^{\circ}}$, at 4.46 μm/s with different sidewalls triangles spacing; (**d**) at $\alpha_{1}{=30}^{{}^{\circ}},\alpha_{2}$ = 30°, S = 0.30 mm and different inlet velocities; (**e**) at $\alpha_{1}=30{}^{\circ}{,\;\alpha }_{2}$ = 50°, S = 0.30 mm, and different inlet velocities; and (**f**) at 4.46 μm/s, S = 0.30 mm, and different $\alpha_{2}$.

**Figure 12 micromachines-13-00338-f012:**
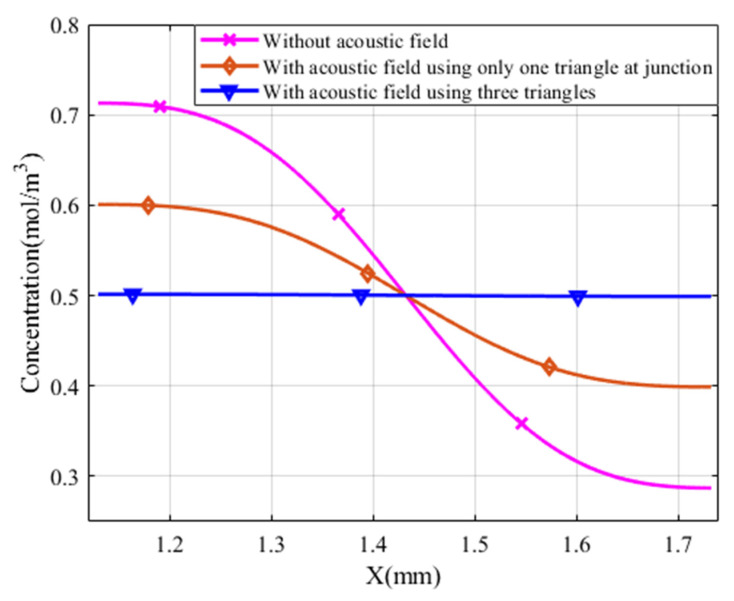
Species concentration profile of inlet velocity 55.6 μm/s at Y = 2.30 mm, ${\alpha =30}^{{}^{\circ}}{,\;\alpha }_{1}{=30}^{{}^{\circ}}{,\;\alpha }_{2\;}{=30}^{{}^{\circ}}$, S = 0.30 mm, and 6 µm oscillation amplitude.

**Figure 13 micromachines-13-00338-f013:**
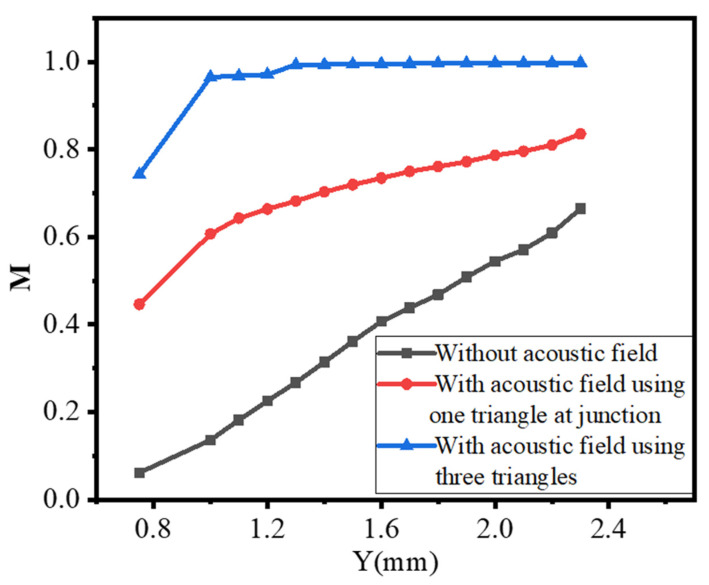
Mixing index comparison with and without acoustic field with inlet velocity 55.6 μm/s ${\alpha =30}^{{}^{\circ}}{,\;\alpha }_{1}{=30}^{{}^{\circ}}{,\;\alpha }_{2\;}{=30}^{{}^{\circ}}$, S = 0.30 mm, and 6 µm oscillation amplitude.

**Table 1 micromachines-13-00338-t001:** Material properties.

Water Properties at T = 25 °C [37]	Value	Units
Viscous dynamics viscosity, µ	890	µPas
$\mathrm{Specific}\;\mathrm{heat}\;\mathrm{capacity},\;C_{p}$	4180	J/kg·K
$\mathrm{Density},\;\rho_{o}$	997	kg/m^3^
$\mathrm{Speed}\;\mathrm{of}\;\mathrm{sound},\;c_{o}$	1497	m/s
$\mathrm{Compressibility},\;k_{o}=1/\left(\rho_{o}c_{o}{}^{2}\right)$	$4.47\times 10^{-10}$	1/Pa
Specific heat capacity ratio, γ	1.012	
$\mathrm{Thermal}\;\mathrm{conductivity},\;k_{th}$	0.61	W/m·K
$\mathrm{Thermal}\;\mathrm{expansion}\;\mathrm{coefficient},\;\alpha =\sqrt{C_{P}\left(\gamma -1\right)/{(\mathrm{Tc}}_{o}{}^{2})}$	$2.74\times 10^{-4}$	1/K
$\mathrm{Thermal}\;\mathrm{diffusivity},\;D_{th}=k_{th}/\left(\rho_{o}c_{p}\right)$	$1.464\times 10^{-7}$	$m^{2}$/s
$\mathrm{Bulk}\;\mathrm{dynamic}\;\mathrm{viscosity},\;\mu_{b}$	2.47	mPas

**Table 2 micromachines-13-00338-t002:** Study input parameters.

Parameters	Values	Units
Inlets velocities	4.46, 8.89, 30, 55.6	µm/s
Oscillation frequency	13	kHz
Oscillation amplitude	3–6	µm
Vertex angles	${\alpha =22}^{{}^{\circ}}{–60}^{{}^{\circ}},\;\alpha_{1}=30^{{}^{\circ}},\;\alpha_{2}{=22}^{{}^{\circ}}{–90}^{{}^{\circ}}$	
Spacing gap S of the side walls triangles	0.15, 0.30, 0.40, 0.60, 0.90	mm
Diffusion coefficient (fluorescein sodium salt)	10^−9^	$m^{2}/s$

## Data Availability

The data that support the findings of this study are available from the corresponding author upon reasonable request.

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
