# Peer review of "A Numerical Investigation of the Mixing Performance in a Y-Junction Microchannel Induced by Acoustic Streaming"

_micromachines, 2022, doi:10.3390/mi13020338_

Round 1
Reviewer 1 Report
I have enjoyed reading this paper. It iii generally well-written and all aspects are well articulated. I have some editorial suggestions:
[1] Some equations are in bold and others are not. Why so?
[2] The overall geometry of the system could have been presented using a more convenient origin for the underlying coordinate system. Referring to Fig. 1a, a possible candidate for this origin is the apex of the 240 degree angle.
[3] In figure 9, 10, and 12 the horizontal axes of the plots is labeled "Width along x direction". If changing the coordinate system's origin is too heavy a burden, the axes label need to be defined in the caption. Something like "position along the x axes in mm measured from the left wall".
[4] The label of the horizontal axes in Figs. 11 and 13 should simply read: position along the y axis. The expression "flow length" is unusual and confusing.
[5] A piece of software like COMSOL Multiphysics has an optimization module. Please rest assured that I am not requesting any new major undertaking. This said, it would seem appropriate to discuss the role of topology optimization in the field of microacoustofluidics in the conclusions. That is, once the physics of the problem has been clearly defined as the authors have done, then the task of optimizing a device can be entrusted to robust optimization schemes.
[6] The use of the English language can be improved here and there. I will only point out a few points that need to be clarified:
- The words "parameters of the" at lines 126 and 127 can be deleted.
- the expression "or convection +I omega" at line 164 is confusing and it can probably be deleted.
- the expression "time periodic component of quantity" at line 166 is confusing. Should the expression simply be "frequency of actuation"?
- The sentence spanning 173 and 174 starting with "Because of thee low" is not grammatically correct and the meaning is unclear. Please clarify.
- The word "phenomenon" on line 240 should be deleted. Also, the expression "difficulty of streaming disturbance" is very confusing. Please rephrase.
- The sentence spanning lines 249-250 has no principal clause and therefore it is incomplete and very confusing.
- Line 258: the plot in Figure 6 does NOT show streaming lines. The plot in question is called a "quiver plot". Please correct.
- The sentence spanning lines 308 and 309 should probably be rephrased as "
Therefore, with only a sharp-edge triangle at the Y junction, mixing is incomplete and fails to achieve the desired objective.
Reviewer 2 Report
Endaylalu et. al. presented a numerical study investigating the mixing performance in a Y-junction microchannel Induced by acoustic Streaming. The effect of different triangle apex angles, heights and distance from each other on mixing performance is studied. Overall, the manuscript is well written, and the results are useful for designing an acoustofluidic mixing device using sidewall triangles. I have few minor comments/questions for the authors.
There are some sentences that should be edited to increase flow and clarity. For example, "in other side S dimensions away from the first sidewall" is not clear. "S dimensions" should be expressed better like "a distance of S away"
Why is the particular frequency selected? In the existing sharp-edge (triangle) based microfluidic devices, the resonant frequency of the transducers is usually between 4 and 6 kHz.
The presented parametric study of the triangle height and apex angle is helpful. The distance S is also useful to evaluate the mixing performance. Did the authors consider/try to change the distance of the first sidewall triangle from the first perpendicular triangle?
What is the limit of mixing performance in terms of inlet flow rate?
Did the authors think about/consider/test a design in which no sidewall triangles are present and only the initial perpendicular triangles mixes the fluids for a range of flow speeds?
